# Extended supercooled storage of red blood cells
Ziya Isiksacan[1,2], Nishaka William[3], Rahime Senturk[1,2,4], Luke Boudreau [1,2], Celine Wooning [1,2,5], Emily Castellanos[1,2,6], Salih Isiksacan[1,2,7], Martin L. Yarmush [1,2,8], Jason P. Acker [3,9] ✉ & O. Berk Usta [1,2] ✉

Red blood cell (RBC) transfusions facilitate many life-saving acute and chronic interventions. Transfusions are enabled through the gold-standard hypothermic storage of RBCs. Today, the demand for RBC units is unfulfilled, partially due to the limited storage time, 6 weeks, in hypothermic storage. This time limit stems from high metabolism-driven storage lesions at +1-6 °C. A recent and promising alternative to hypothermic storage is the supercooled storage of RBCs at subzero temperatures, pioneered by our group. Here, we report on long-term supercooled storage of human RBCs at physiological hematocrit levels for up to 23 weeks. Specifically, we assess hypothermic RBC additive solutions for their ability to sustain supercooled storage. We find that a commercially formulated next-generation solution (Erythro-Sol 5) enables the best storage performance and can form the basis for further improvements to supercooled storage. Our analyses indicate that oxidative stress is a prominent time- and temperature-dependent injury during supercooled storage. Thus, we report on improved supercooled storage of RBCs at −5 °C by supplementing Erythro-Sol 5 with the exogenous antioxidants, resveratrol, serotonin, melatonin, and Trolox. Overall, this study shows the long-term preservation potential of supercooled storage of RBCs and establishes a foundation for further improvement toward clinical translation.

The need for red blood cell (RBC) transfusion is steadily growing, with 36 thousand units transfused daily in the United States alone[1,2]. Despite the yearly donation of 120 million blood units, the global demand has never been fulfilled, especially during times like pandemics or wars when the supply drops dramatically. In such worldwide shortages as experienced recently, the waste of even a single RBC unit before finding a recipient is intolerable[3,4]. To this end, blood banks play a strategic role in the maintenance of adequate RBC supply by ensuring that the RBC units are held at hypothermic storage (+1-6 °C) for ∼ 42 days, the gold standard RBC storage practice[5]. Cryopreservation of RBCs is an alternative storage method but is mostly favored for storing rare blood types or maintenance of strategic stockpiles, where the concerns over meeting immediate demand or in times of massive transfusion make this approach impractical[6].

RBC units stored at hypothermic conditions experience biochemical, metabolic, and structural damages referred to as storage lesions[7–9]. These involve a gradual rise in hemolysis, both intra and extracellular acidification, accumulation of potassium in the extracellular space, rigidity, and depletion of redox reserves[10–14]. The magnitude of the damage depends significantly on blood processing, storage time, lifestyle, and donor genotype/phenotype factors[3]. Overall, the accumulation of damages lowers the quality of stored RBCs. Eventually, the decline in cell quality can diminish transfusion efficacy and safety[15–17], especially for chronic transfusion recipients and vulnerable pediatric or cancer patients[18,19]. Consequently, the quality outcomes of the RBC storage via the gold standard method has been questioned by multiple organizations, including the 2008 retrospective clinical study and the 2022 US National Heart, Lung, and Blood Institute (NHLBI) symposium[20,21].

Over the years, the limitations in hypothermic storage of RBCs have been addressed through the development of new additive solutions (AS) which replaced storage in saline/glucose solutions[22]. First, an additive

[1]Center for Engineering in Medicine and Surgery, Massachusetts General Hospital, Harvard Medical School, Boston, MA, USA. [2]Shriners Children's, Boston, MA, USA. [3]Department of Laboratory Medicine and Pathology, University of Alberta, Edmonton, AB, Canada. [4]Department of Chemical Engineering, Eindhoven University of Technology, Eindhoven, Netherlands. [5]Department of Human Biology, Scripps College, Claremont, CA, USA. [6]Department of Psychology, Amherst College, Amherst, MA, USA. [7]Department of Electrical-Electronics Engineering, Bilkent University, Ankara, Turkey. [8]Department of Biomedical Engineering, Rutgers University, Piscataway, NJ, USA. [9]Innovation and Portfolio Management, Canadian Blood Services, Edmonton, AB, Canada. ✉e-mail: jacker@ualberta.ca; ousta@mgh.harvard.edu

solution with saline, adenine, glucose, and mannitol (SAGM) was formulated in 1981, followed by AS-1, AS-3, and AS-5[23–26]. These earlier additive solutions were acidic (pH: 5-6), and the gradual accumulation of lactic acid at +1-6 °C further increased the intracellular and extracellular acidity in RBC units, which inhibits glycolysis. The inhibition of glycolysis depletes the adenosine triphosphate (ATP) reserves and diminishes the function of RBCs. A later revision to SAGM, thus, included guanosine (PAGGSM) to maintain ATP levels during storage[27]. The next-generation additive solutions like Erythro-Sol 5 (E-Sol 5), SOLX (AS-7), and phosphate-adenine-glucose-guanosine-gluconate-mannitol (PAG3M) were designed to have an alkaline starting pH. Additionally, they do not contain chloride, which helps promote chloride efflux from the RBCs into the solution especially in the first 7 days of storage and counterflow of hydroxyl anions from the solution into the RBCs (chloride shift) and promotes an alkaline intracellular pH during the storage[28–30]. These recent solutions also have higher pH buffer capacities due to the inclusion of phosphate (PAGGSM, E-Sol 5, AS-7, PAG3M) and bicarbonate (AS-7)[31].

Despite the improved metabolism of RBCs stored in these next-generation solutions, the accumulation of reactive oxygen species (ROS) remains a hurdle during hypothermic storage[32–34]. Some common antioxidants like vitamin C and N-acetylcysteine have been used to alleviate ROS in RBC units for better storage and post-transfusion recovery[35–37]. Other antioxidants, including uric acid, resveratrol, or quercetin, have also been studied to improve quality of stored RBCs. However, these antioxidants have not been included in commercial RBC additive solution formulations, yet[38–40]. Unlike the RBC additive solutions, antioxidants have been featured in various commercially available storage mediums[41]. For example, HypoThermosol-FRS, a hypothermic cell, tissue, and organ preservation medium, includes Trolox, a potent general antioxidant, in its formulation[42]. Similarly, the University of Wisconsin (UW) solution includes glutathione, an important endogenous antioxidant[43]. Given that antioxidants are already in storage mediums formulated for other cells and tissues, there is still room for developing improved RBC additive solutions supplemented with effective antioxidants. However, improvements in hypothermic storage of RBC units through the reformulation of additive solutions might not satisfactorily address all the RBC shortage and storage issues that we experience today.

To this end, supercooled storage of RBCs provides a novel alternative to hypothermic storage. A supercooled liquid exists in a metastable state where the liquid is cooled below its melting point ($T_m$) without forming ice crystals[44–47]. In the supercooled state, ice crystallization occurs once the temperature reaches the homogenous nucleation temperature ($T_h$) or a heterogenous nucleation site triggers the nucleation between $T_m$ and $T_h$. In nature, species like fish, insects, and amphibians can attain supercooled states during subzero winters and still survive[48]. Learning from these species, preservation of living organisms in a supercooled state can enable deeper metabolic stasis compared to hypothermic storage and prolong preservation times. This can be performed while warranting that organisms do not form ice below $T_m$ and hence are protected from detrimental effects of ice crystallization. RBC supercooled storage strategies, which center around significantly slowing down (relative to hypothermic storage) − but not stopping (as in cryopreservation) − the cellular metabolism, could improve cell quality and prolong storage time. Indeed, deeper metabolic stasis through supercooled storage has previously been achieved for bacterial and yeast cells, peripheral blood stem cells, turkey spermatozoa, and hepatocytes[42,49]. At the organ level, rodent and human livers were supercooled without freezing at our center[50]. Supercooled storage of RBCs was demonstrated by our research group at low hematocrit levels ( <0.001%)[51]. RBCs were supercooled by ensuring the sealing of the sample-air interface, the most likely site of heterogeneous ice nucleation, with immiscible hydrocarbon-based liquids. In that study, AS-3 and UW solutions were used for RBC suspension. In a recent follow-up study, we investigated whether the quality of stored RBCs was improved at −4 °C relative to +4 °C in the next-generation PAG3M solution in the presence of two extracellular additives: trehalose and polyethylene glycol (PEG)[52]. Despite the

improvements with PEG addition into the PAG3M formulation, the samples stored in PAG3M showed much higher hemolysis levels at −4 °C compared to +4 °C beyond 6 weeks of storage.

Here, we aimed to find the best performing commercially formulated hypothermic RBC additive solution for use in long-term supercooled storage. Specifically, we investigated the suitability of two next-generation (E-Sol 5 and AS-7) and two conventional additive solutions (CPDA-1 and SAGM). The interface between the RBC suspensions and air within the sample tubes was the most likely site of heterogenous ice nucleation during supercooling. Hence, ice nucleation was hindered by sealing this interface using mineral oil (see Fig. 1e and Methods). Among all groups, the samples stored in E-Sol 5 experienced the lowest hemolysis at both −5 and +4 °C during 10 weeks of storage. While hemolysis at +4 °C was lower than at −5 °C at earlier time points, longer-term – up to 23 weeks – experiments showed that supercooled storage became significantly advantageous beyond 20 weeks. Overall, decreased lactate levels at subzero temperatures demonstrated the suppression of metabolism relative to hypothermic storage. We observed that the supercooled samples experienced hemolysis levels higher than the Food and Drug Administration (FDA)-approved 1% threshold. Also, the samples stored at subzero temperatures experienced time-dependent increase in lipid peroxidation and decrease in antioxidant capacity. We hypothesized that these oxidative injuries, predominantly accumulating at earlier time points, can be alleviated by exogenous antioxidant supplementation. Therefore, we reformulated E-Sol 5 to include either of the four antioxidants: resveratrol, serotonin, melatonin, and Trolox at 100 μM concentrations. All of the antioxidants demonstrated suppressed hemolysis during 10 weeks of supercooled storage at −5 °C. Notably, hemolysis levels for the samples supplemented with serotonin, melatonin, or Trolox stayed below the 1% threshold after 6 weeks of storage at −5 °C[53].

## Results and discussion
### Comparing additive solutions for supercooled storage

The experimental steps for the supercooled storage of RBCs are illustrated in Fig. 1. We initially screened 6 additive solutions: E-Sol 5, AS-7, PAG3M, CPDA-1, SAGM, and 5% trehalose-supplemented UW (UW+Tre). Although UW solution is not used for RBCs, it was chosen to test the suitability of a common hypothermic storage medium for the supercooled storage of RBCs in the presence of a known membrane protectant, trehalose. Firstly, we suspended RBCs in these solutions and stored them at +4 °C and −5 °C for 4 weeks (Supplementary Fig. 1). The samples stored in E-Sol 5, AS-7, PAG3M, and CPDA-1 experienced similar hemolysis levels of 1–2% at −5 °C. In contrast, both SAGM and UW+Tre caused more than 5% hemolysis at −5 °C. Hypothermic storage in all the solutions resulted in ~ 1% hemolysis, except UW+Tre, which caused ~ 12% hemolysis and was eliminated from further testing. PAG3M was also eliminated due to its poor performance at −5 °C relative to +4 °C beyond 6 weeks[52].

Following this initial screening, we assessed the supercooled storage of two next-generation additive solutions, E-Sol 5 and AS-7, together with CPDA-1, a solution used in the United States, and SAGM, the standard in Europe and Canada (Table 1). Specifically, these solutions were compared in terms of hemolysis, metabolic activity, and lipid peroxidation. RBC samples were prepared in either of the solutions and stored at hypothermic (+4 °C) and supercooled (−5 °C) conditions. Hemolysis was measured for all the samples at week 6 (Fig. 2a) and week 10 (Fig. 2b) time points. After 6 weeks at +4 °C, the samples stored in E-Sol 5 experienced the lowest hemolysis compared to the samples stored in AS-7, CPDA-1 and SAGM (Supplementary Fig. 3a). Similarly, hemolysis for the supercooled samples stored in E-Sol 5 was significantly lower compared to those in AS-7, CPDA-1 and SAGM at −5 °C (Fig. 2a). Although the difference in hemolysis levels across the two temperatures were significant for the samples stored in E-Sol 5, the statistically significant differences across the two temperatures for all the other solutions were much higher (Fig. 2a). The difference in hemolysis between hypothermic and supercooled storage was especially clear for SAGM.

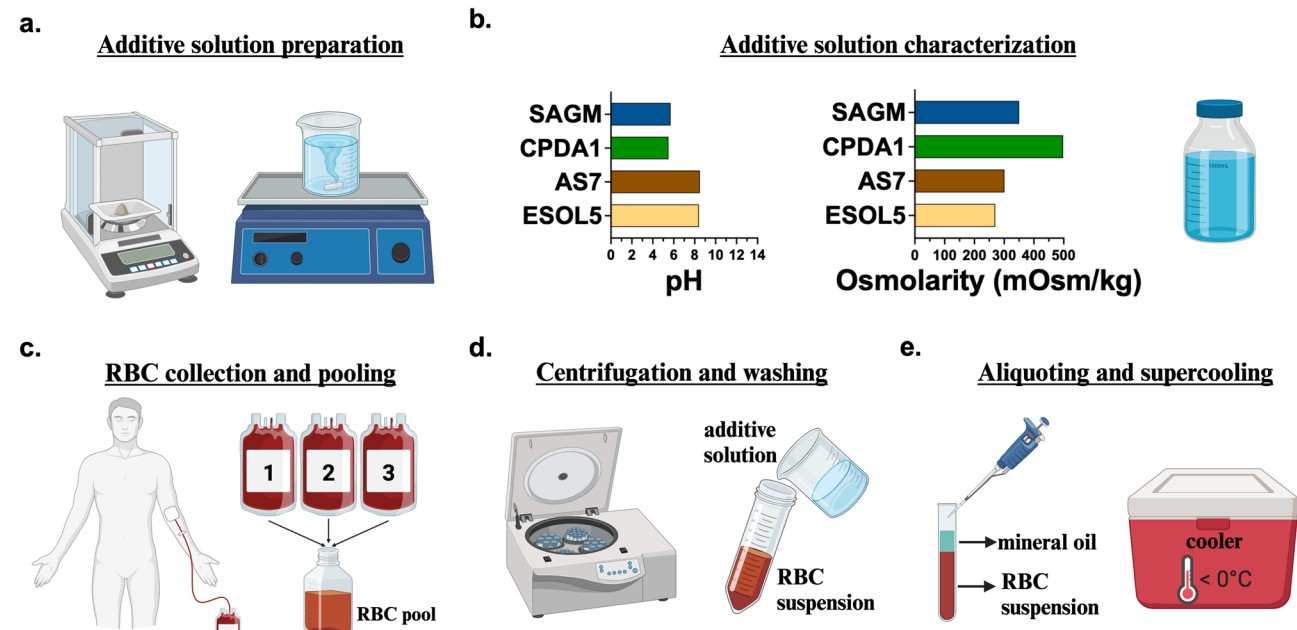

**Fig. 1 | Schematic of experimental steps for the supercooled storage of red blood cells (RBCs). a** Additive solutions were prepared fresh in-house in sterile conditions. **b** Additive solutions were characterized for pH and osmolarity. **c** A single experiment was performed from a pool of packed RBCs from 3 donors. **d** Pools were washed and resuspended in their respective additive solutions. **e** Samples were aliquoted as 1 mL units at ∼ 50% hematocrit levels and sealed with a 0.5 mL mineral oil layer to prevent stochastic freezing at subzero temperatures. Samples were stored in coolers (Engel MHD-13, Engel, FL, USA) set at designated subzero temperatures. The figure was prepared in Biorender.com.

After 6 weeks of storage, E-Sol 5 was the best performing additive solution in terms of hemolysis at both temperatures. From week 6 to week 10, hemolysis levels increased in each group. At week 10, hemolysis at +4 °C was significantly higher for the samples stored in AS-7, CPDA-1 and SAGM compared to the samples stored in E-Sol 5 (Supplementary Fig. 3b). Similarly, hemolysis for the samples stored in E-Sol 5 at −5 °C was significantly lower compared to all the other solutions (Fig. 2b). We observed that hemolysis for the samples stored in E-Sol 5 at +4 °C was lower than those stored at −5 °C. Yet, for the samples stored in the other solutions, the storage at −5 °C resulted in substantially much higher hemolysis compared to the storage at +4 °C (Fig. 2b). All the samples were then visually assessed at week 21, and we observed that the samples stored in CPDA-1 and SAGM experienced complete hemolysis at both temperatures (Supplementary Fig. 4). Overall, these results indicated that E-Sol 5 was the best commercially developed additive solution for which RBCs experienced the lowest hemolysis during supercooled storage.

Glycolytic activity, as quantified by lactate production, was measured for the RBC samples after 6 weeks of storage (Fig. 2c and Supplementary Fig. 3c). Increased lactate levels, compared to the mean initial lactate concentration of 4.66 mmol/L, demonstrated the continuation of glycolytic activity. The lactate levels increased significantly more for the samples stored in E-Sol 5 and AS-7 at both temperatures than the samples stored in CPDA-1 and SAGM (Supplementary Fig. 3c). At both temperatures, the lactate levels for the samples stored in E-Sol 5 were higher than the levels for the samples stored in the other solutions at the corresponding temperatures. The lactate levels for the samples stored in E-Sol 5 and AS-7 were lower at −5 °C than at +4 °C, indicating the suppression of metabolism with the decreasing temperature as expected (Fig. 2c). Nevertheless, we did not measure a similar depression of metabolism for the samples stored in CPDA-1 and SAGM for which the metabolic activity was already significantly lower at +4 °C compared to E-Sol 5 and AS-7. It is important to note that the higher lactate levels for the samples stored in E-Sol 5 and AS-7 cannot be explained by their initial glucose levels. While E-Sol 5 and AS-7 formulations contained higher glucose concentrations than SAGM, CPDA-1 had the highest starting glucose concentration. Despite the highest initial glucose concentration in CPDA-1, the samples stored in CPDA-1 generated significantly lower lactate levels at both temperatures than the samples stored in E-Sol 5 and AS-7 (Supplementary Fig. 3c). Therefore, the lower lactate levels for the samples stored in CPDA-1 and SAGM were likely driven by the acidity (pH: 5-6) of these solutions. During storage, the accumulation of lactic acid increased the acidity for all the groups. However, the reduction in pH for CPDA-1 and SAGM becomes faster and more significant. This is corroborated by the fact that the samples stored in these solutions experienced higher hemolysis over time (Fig. 2a).

Thiobarbituric Acid Reactive Substances (TBARS) concentration, a measure of lipid peroxidation, increased significantly in all groups during storage compared to the initial levels. Yet, the statistical difference in TBARS levels across different groups were insignificant at week 6 (Fig. 2d). Hence, neither TBARS nor the initial pH (Fig. 1b) levels of the solutions alone explained why the samples stored in SAGM experienced the highest hemolysis at −5 °C (Figs. 2a, b). All the solutions except SAGM had two common properties that might play significant roles in the resulting hemolysis levels. First, E-Sol 5, AS-7, and CPDA-1 had higher pH buffering capacities than SAGM (Table 1). Second, these solutions were chloride-free. We, thus, hypothesized that the absence of chloride in the other solutions likely promoted chloride shift to better maintain the glycolytic activity compared to SAGM.

Storage in SAGM resulted in slight swelling of the RBCs as indicated by an increased mean cell volume (MCV) especially at −5 °C compared to the initial level (Supplementary Fig. 5). At week 6, the statistical difference in MCV levels for the samples stored in each solution across +4 °C and −5 °C was insignificant. The mean corpuscular hemoglobin concentration (MCHC) levels for the samples stored in SAGM were lower than the other solutions at both temperatures due to the swelling of RBCs in SAGM (Fig. 3e and Supplementary Fig. 3d). The MCHC levels for the samples stored in SAGM were significantly lower than the initial MCHC level (Supplementary Fig. 5b). The MCHC levels for the samples stored in E-Sol 5, AS-7, and CPDA-1 were significantly higher than the respective initial MCHC levels. The statistical differences in MCHC levels for the samples stored in each temperature were insignificant across +4 °C and −5 °C at week 6 except

SAGM (Fig. 2e). For all the groups, the mean cellular hemoglobin (MCH) levels were retained compared to the initial levels and remained the same at week 6 (Fig. 2f). This suggested that the stored RBCs conserved their hemoglobin contents independent of the storage temperature and additive solution.

### Injury assessment during long-term supercooled storage in E-Sol 5

It was shown in Fig. 2b that the samples stored in E-Sol 5 resulted in the lowest hemolysis during 10 weeks of storage at −5 °C. Thus, we performed long-term supercooled storage of RBCs in E-Sol 5 at +4 °C, −5 °C, and −8 °C for up to 23 weeks. Hemolysis data up to week 10 are shown in Fig. 3a. At week 3, hemolysis at +4 °C was significantly lower than at the other temperatures, and the hemolysis at −8 °C was the highest. At week 7, the

hemolysis at +4 °C, −5 °C, and −8 °C increased compared to the hemolysis levels at the respective temperatures at week 3 (Supplementary Fig. 6a). At week 7, hemolysis at +4 °C was lower than at −5 °C and −8 °C, and hemolysis at −8 °C was again significantly higher than at −5 °C. We did not observe any statistical significance between week 7 and week 10 at +4 °C. This was also the case at −5 °C (Supplementary Fig. 6a) 8. Instead, the statistical significance was observed between week 3 and week 10 at these two temperatures. Hemolysis at week 10 at +4 °C was lower than −5 °C, similar to the comparison of the two temperatures at week 7. At week 10, the hemolysis at −8 °C was measured as more than 10%, significantly higher than the other temperatures.

We also quantified the glycolytic activity by measuring the lactate levels at these different temperatures (Fig. 3b). Compared to the starting level, the lactate levels at all the temperatures increased during storage. At week 3, the lactate level for the samples stored at +4 °C was significantly higher than those at −5 °C and −8 °C. We did not observe any statistical difference in lactate levels between the two subzero temperatures. From week 3 to week 7, the lactate levels increased significantly for the samples stored at −5 °C and −8 °C (Supplementary Fig. 6b). At week 7, the lactate level for the samples stored at +4 °C was higher than those at −5 °C and −8 °C. Again, we did not measure any statistical difference between the samples stored at the two subzero temperatures. When the storage time reached week 10, the lactate level for the samples stored at +4 °C was above the maximum concentration reportable by our instrument, indicating that the lactate level raised even further. The lactate level at −5 °C increased significantly from week 3 to week 10 suggesting the continuation of glycolytic activity.

We quantified lipid peroxidation (TBARS) and total antioxidant capacity (TAC) at all time points, to understand the oxidative stress related injury mechanisms during supercooled storage of RBCs (Fig. 3c, d). Compared to the initial mean TBARS of 1.1 mM, all samples underwent significant lipid peroxidation during the first 3 weeks of storage (Fig. 3c). At week 3, a statistically significant difference in TBARS levels was observed

### Table 1 | Composition of red blood cell additive solutions

| Constituents (mmol/L) | E-Sol 5 | AS-7 | CPDA-1 | SAGM |
|---|---|---|---|---|
| Sodium chloride | 0 | 0 | 0 | 150 |
| Sodium bicarbonate | 0 | 26 | 0 | 0 |
| Sodium phosphate dibasic | 20 | 12 | 0 | 0 |
| Sodium phosphate monobasic | 0 | 0 | 18.5 | 0 |
| Citric acid | 0 | 0 | 15.5 | 0 |
| Mannitol | 41 | 55 | 0 | 30 |
| Glucose | 111 | 80 | 176.7 | 45 |
| Sodium citrate | 25 | 0 | 89.3 | 0 |
| Adenine | 2 | 2 | 0.4 | 1.25 |
| pH | 8.4 | 8.5 | 5.5 | 5.7 |
| Osmolarity (mOsm/kg) | 270 | 203 | 498 | 350 |

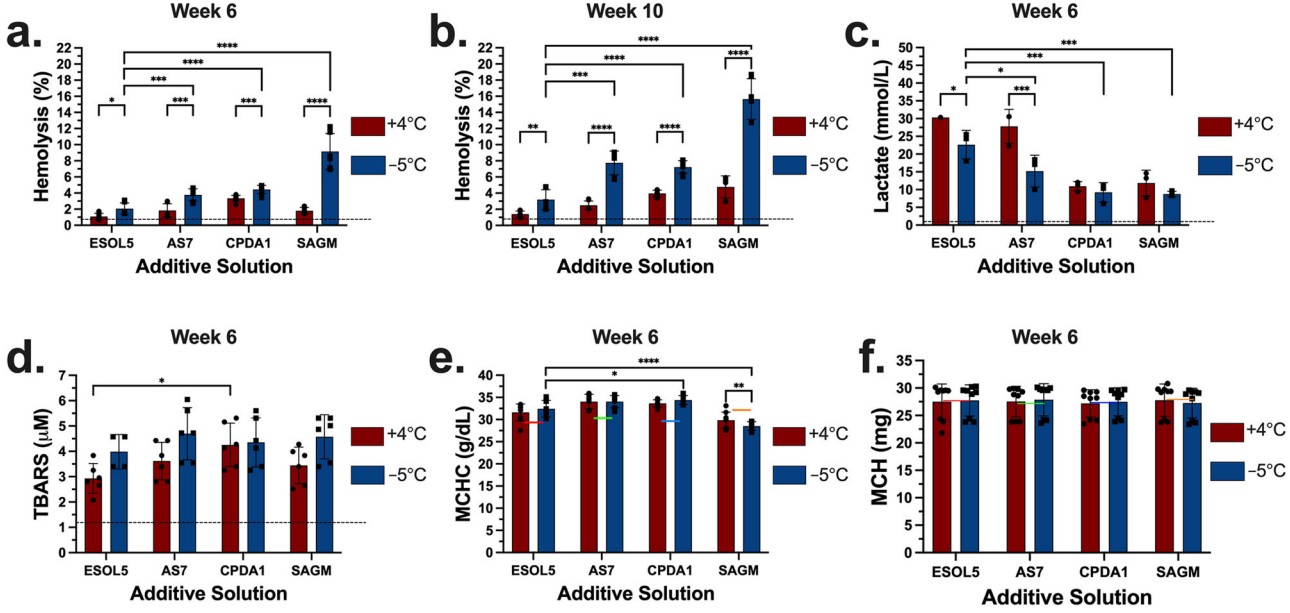

**Fig. 2 | Comparison of E-Sol 5, AS-7, CPDA-1, and SAGM for hemolysis, hemolysis-corrected lactate, TBARS, MCHC, and MCH at +4 °C and −5 °C after 6 and 10 weeks of storage. a** The samples were stored for 6 weeks, and hemolysis was measured. **b** The samples were stored for 10 weeks, and hemolysis was measured. **c** Data shows lactate level corrected for hemolysis after 6 weeks. Measured total lactate level was normalized by fraction of unlysed RBCs. **d** Data shows TBARS levels after 6 weeks. **e** Data shows MCHC levels at both temperatures after 6 weeks. **f** Data shows MCH levels at both temperatures after 6 weeks. Colored and dashed lines represent the mean day 1 levels following 3x washing, overnight storage at +4 °C, and final 1x washing in the respective solutions. Data represent mean ± standard

deviation from 3 biological replicates (N = 3) and three technical replicates (n = 3, n = 1-3 for lactate, n = 2 for TBARS) for each biological replicate. Each biological replicate was from a pool of 3 donor samples. A two-way analysis of variance (ANOVA) followed by Tukey's *post hoc* test was performed to evaluate significant differences between conditions: *p < 0.05; **p < 0.01; ***p < 0.001; ****p < 0.0001. Comparisons were shown across the two temperatures for each solution and for E-Sol 5 against the other solutions. TBARS Thiobarbituric Acid Reactive Substances. MCHC mean cell hemoglobin concentration. MCH mean corpuscular hemoglobin. See Supplementary Fig. 3 for comparisons across the solutions at each temperature.

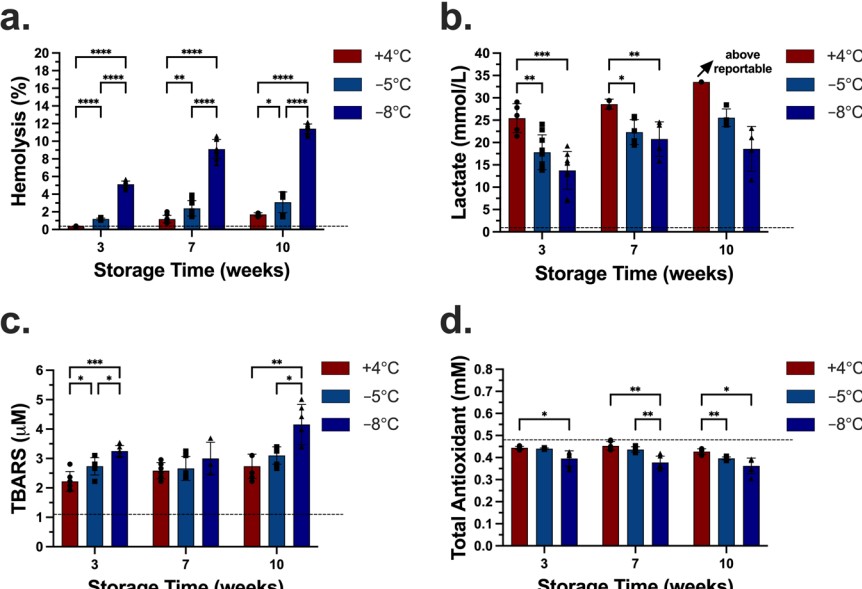

**Fig. 3 | Comparison of storage performance in E-Sol 5 at 3 different temperatures after 3, 7, and 10 weeks of storage. a** Hemolysis increased with increasing time of storage and decreasing temperature. **b** Data shows lactate levels normalized by fraction of unlysed red blood cells after 3, 7, and 10 weeks. Measured total lactate level was normalized by percent of unlysed RBCs. **c** Data shows TBARS levels after 3, 7, and 10 weeks. **d** Data shows TAS levels after 3, 7, and 10 weeks. Dashed lines represent the mean day 1 levels following 3x washing, overnight storage at +4 °C, and final 1x washing in E-Sol 5. Data represent mean ± standard deviation from 3 biological replicates ($N = 3$) and 1-3 technical replicates for each biological replicate ($n = 3$ for hemolysis, $n = 1$-3 for lactate, $n = 2$ for TBARS and TAS). Each biological replicate was from a pool of 3 donor samples. A two-way analysis of variance (ANOVA) followed by Tukey's *post hoc* test was performed to evaluate significant differences between conditions: *$p < 0.05$; **$p < 0.01$; ***$p < 0.001$; ****$p < 0.0001$. Comparisons were shown across the three temperatures at each time point. See Supplementary Fig. 6 for comparisons across the time points at each temperature. TBARS Thiobarbituric Acid Reactive Substances. TAC Total Antioxidant Capacity.

across all the groups, and the TBARS level was the lowest for the samples stored at +4 °C. At week 10, the TBARS level in the samples stored at −8 °C was significantly higher than the samples stored at +4 °C and −5 °C. We observed no statistical difference in TBARS levels between the samples stored at +4 °C and −5 °C at week 7 and week 10. The TBARS levels in the samples stored at +4 °C increased significantly from week 3 to week 7 and from week 3 to week 10 (Supplementary Fig. 6c). At −5 °C, we observed a significant increase in TBARS levels only from week 3 to week 10 (Supplementary Fig. 6c). The TBARS levels in the samples stored at −8 °C increased significantly from week 7 to week 10. Despite the increase in TBARS levels at different temperatures for week 3 onward, the majority of lipid peroxidation occurred earlier from day 1 to week 3 at all the temperatures (Fig. 3c).

Lipid peroxidation is only one of the oxidative stress injury mechanisms and hence does not provide a complete picture of such injuries. Thus, here, we supplemented the TBARS measurements with the total antioxidant capacity (TAC) assays of the stored samples to provide additional insight on oxidative stress during storage (Fig. 3d). TAC was quantified based on the ability of the endogenous antioxidants of the stored RBCs to inhibit the oxidation of the externally added 2,2'-Azino-di-(3-ethylbenzthiazoline sulphonate) by metmyoglobin. The initial TAC level was measured as 0.48 mM. We observed that the TAC levels decreased during storage at all the temperatures. The statistical differences in the TAC levels between the samples stored at +4 °C and −5 °C were significant at week 10. This suggested that antioxidants were depleted more rapidly for the samples stored at −5 °C than the samples stored at +4 °C. The TAC levels for the samples stored at −8 °C were depleted more significantly than the samples stored at +4 °C at all time points, and at −5 °C at week 7. More specifically for the samples stored at −5 °C, the TAC levels decreased significantly from week 3 to week 10 and from week 7 to week 10 (Supplementary Fig. 6).

Following quantitative assays at multiple time points during 10 weeks of storage, more samples were kept at their respective temperatures and visually assessed from their supernatant colors over the weeks. We aimed to find a storage time where the hemolysis for the samples stored at +4 °C

became significantly higher than the hemolysis for the samples stored at −5 °C. At week 21, we observed more redness in the supernatants of the samples stored at +4 °C implying an elevated hemoglobin release from RBCs compared to −5 °C (Fig. 4). We hypothesize that the ATP reserves for the samples stored at +4 °C were significantly depleted during the long-term storage compared to the metabolically suppressed samples at −5 °C. Notably, the hemolysis for the samples stored at +4 °C was 6.14%, which was significantly higher than the hemolysis for the samples stored at −5 °C (4.08%). During the two consecutive weeks, similar significant differences in hemolysis were observed between the two temperatures. At week 23, the hemolysis for the samples stored at +4 °C was 7.06%, which was significantly higher than the +4 °C level at week 22 (Supplementary Fig. 7). Yet, the increase in hemolysis for the samples stored at −5 °C was insignificant from week 21 to week 22 or from week 22 to week 23.

**Antihemolytic effects of antioxidant supplementation in super-cooled storage**

Despite the significantly lower hemolysis at −5 °C beyond the cutoff storage time (Fig. 4), the quantification of TBARS and total antioxidant capacity at earlier time points (Fig. 3d) indicated that endogenous antioxidants were depleted in a temperature- and time-dependent manner during supercooled storage. Thus, to further improve supercooled storage in E-Sol 5, we assessed its success when supplemented with either of four exogenous antioxidants, resveratrol, serotonin, melatonin, and Trolox. To this end, we, first, used a rapid oxidative stress injury model using an oxidative stress inducer, cumene hydroperoxide (CumOOH), to assess the most effective concentrations for these antioxidants[54]. We incubated 1% hematocrit RBCs with 500 μM CumOOH in the presence of 50 or 100 μM of these antioxidants. This study suggested that 100 μM was the more effective concentration to reduce hemolysis (Supplementary Fig. 8).

We, thus, performed supercooled storage of RBCs at −5 °C in E-Sol 5, either not supplemented (control) or supplemented with one of these antioxidants at 100 μM concentration. All the samples were supercooled at −5 °C for 10 weeks. Hemolysis levels were measured at week 6 (Fig. 5a) and

week 10 (Fig. 5b) time points as a quality marker relative to the antioxidant-unsupplemented samples stored at +4 °C. The initial mean hemolysis was 0.4%. At week 6, the hemolysis for the samples stored at +4 °C increased to 0.57%. The hemolysis for the samples stored at −5 °C for the control group was 1.22%, which was above the FDA-approved 1% threshold for 6 weeks of storage. All antioxidants, except resveratrol, significantly reduced the hemolysis compared to the control group and facilitated the hemolysis below 1% at week 6. At week 10, the hemolysis for the samples stored at +4 °C increased to 0.86%. All antioxidants decreased the hemolysis levels relative to the control, which was measured as 1.94%. Specifically, we achieved 1.73, 1.52, 1.30, and 1.38% hemolysis for resveratrol, serotonin, melatonin, and Trolox, respectively.

Time- and temperature-dependent increase in hemolysis levels during supercooled storage in E-Sol 5 (Fig. 3a) was complemented with the increase in the TBARS levels (Fig. 3c) and the depletion of the total antioxidant capacity of the supercooled samples (Fig. 3d). These findings led us to the use of exogenous antioxidants to reduce hemolysis −5 °C. We chose these four antioxidants that had demonstrated antihemolytic activity in RBCs. For example, resveratrol is a non-flavonoid compound with three phenolic groups. It acts as a free radical scavenger through the lipid bilayer by transferring the proton from its phenolic group to the free radicals[55]. Although not as effective as the other antioxidants, resveratrol still reduced hemolysis during supercooled storage. Serotonin is a neurotransmitter, yet it has been demonstrated to also have potent antioxidant activity. Serotonin binds to lipids on both alkyl chains and locates at the hydrophobic-hydrophilic interface of RBCs, allowing it to alleviate ROS that would otherwise cause membrane oxidation[56]. Melatonin is a hormone synthesized from serotonin and a natural antioxidant. Its antioxidant capacity lies in the indole moiety of the molecule, which can scavenge different types of radicals (hydroxyl, alkoxyl, and peroxyl). Previous studies demonstrated that melatonin shows lipoperoxyl radical-scavenging activity and delays membrane protein degradation and precipitation of hemin onto the RBC membrane[54]. Trolox is a synthetic analog of Vitamin E and a chain-breaking antioxidant. It was proven as a hydroxyl and alkoxyl radical scavenger in aqueous and lipid environments and a peroxyl radical scavenger in aqueous solution[57]. It is evident from the previous findings that all these antioxidants possess different molecular/biochemical structures and undertake distinct mechanisms of action to alleviate oxidative stress. We demonstrated that antioxidant supplementation helped restore the depleted antioxidant capacity and reduced hemolysis during 10 weeks of supercooled storage. Overall, we posit that a holistic mapping of the supercooling-associated injury mechanisms is necessary to more effectively enhance short-term and long-term supercooled storage through more targeted strategies.

## Conclusion

We reported on improved supercooled storage of human red blood cells (RBC) at physiological hematocrit levels for up to 23 weeks as an alternative to 6 weeks of hypothermic storage. We assessed two next-generation additive solutions, E-Sol 5 and AS-7, together with two conventional counterparts, CPDA-1 and SAGM, for their suitability to provide improved

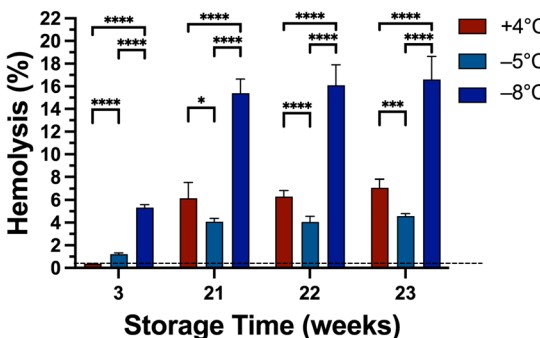
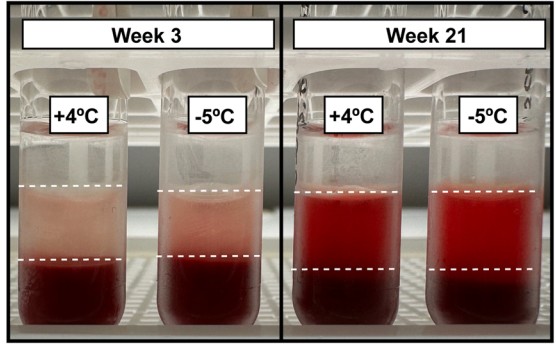

**Fig. 4 | Long-term comparison of storage performance in E-Sol 5 at +4 °C and −5 °C after 3, 21, 22, and 23 weeks of storage.** Samples were assessed for their supernatant colors where redness is proportional to hemolysis. At week 21 and later time points, the hemolysis for the samples stored at +4 °C was significantly higher than the hemolysis for the samples stored at −5 °C. Black dashed line represents the mean day 1 level following 3x washing, overnight storage at +4 °C, and final 1x washing in E-Sol 5. Data represent mean ± standard deviation from 2 biological replicates (N = 2) and 3 technical replicates (n = 3) for each biological replicate. Each biological replicate was from a pool of 3 donor samples. A two-way analysis of variance (ANOVA) followed by Tukey's *post hoc* test was performed to evaluate significant differences between conditions: *$p < 0.05$; **$p < 0.01$; ***$p < 0.001$; ****$p < 0.0001$. Comparisons were shown across the three temperatures at each time point. See Supplementary Fig. 7 for comparisons across the time points at each temperature.

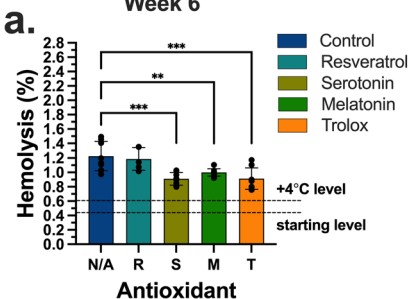
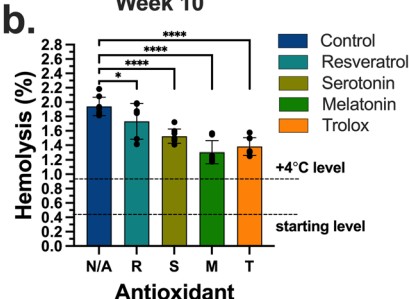

**Fig. 5 | Comparison of antihemolytic effects of 4 antioxidants during supercooled (−5 °C) storage after 6 and 10 weeks of storage. a** The samples were stored for 6 weeks at −5 °C. Serotonin, melatonin, and Trolox decreased the hemolysis below 1%. **b** The samples were stored for 10 weeks at −5 °C. All antioxidants decreased the hemolysis relative to the antioxidant-unsupplemented case at −5 °C (control, N/A). Data represent mean ± standard deviation from 3 biological replicates (N = 3) and 2 or 3 technical replicates (n = 2-3) for each biological replicate. Each biological replicate was from a pool of 3 donor samples. A one-way analysis of variance (ANOVA) followed by Dunnett's *post hoc* test was performed to evaluate significant differences between conditions: *$p < 0.05$; **$p < 0.01$; ***$p < 0.001$; ****$p < 0.0001$. For clarity, comparisons were made for the control (N/A) against the antioxidant cases at −5 °C.

and extended supercooled storage. All the supercooled samples exceeded the Food and Drug Administration (FDA)-approved 1% hemolysis level standardized for up to 6 weeks of hypothermic storage. Yet, compared to the other solutions, the samples stored in E-Sol 5 experienced lower hemolysis and ensured continued glycolytic activity during storage at −5 °C. While hypothermic storage led to lower hemolysis than supercooled storage at earlier time points, the supercooled samples experienced lower hemolysis than the samples stored in hypothermic conditions at and beyond week 21. Oxidative stress was an important mechanism of injury at subzero temperatures. To this end, we recruited four exogenous antioxidants, resveratrol, serotonin, melatonin, and Trolox, to decrease hemolysis levels at −5 °C. Through the supplementation of these antioxidants in E-Sol 5, we kept the hemolysis levels significantly lower than the antioxidant-unsupplemented samples during 10 weeks of supercooled storage. Supplementation of serotonin, melatonin, or Trolox in E-Sol 5 kept the hemolysis below the FDA-approved level during the first 6 weeks of supercooled storage. This study demonstrated that E-Sol 5 is the best commercially formulated RBC hypothermic additive solution for supercooled storage and its performance can further be enhanced by exogenous antioxidant supplementation to meet FDA-mandated criteria for transfusion of stored RBC products after 6 weeks of storage.

## Methods

### Inclusion and ethics statement

All research complies with all relevant ethical regulations, and the experimental protocol was approved by the Institutional Review Board of Massachusetts General Hospital (protocol number: 2019P003498). Studies were conducted in the Biosafety Level 2 laboratories in the same building.

### Experimental materials

Commercial reagents and antioxidants were purchased from Sigma-Aldrich (MA, USA). Round-bottom polypropylene centrifuge tubes and polystyrene serological pipets were purchased from Genesee Scientific (CA, USA). Clear flat-bottom polystyrene 96-well plates and Falcon 5 mL round-bottom polystyrene tubes were purchased from Corning (MA, USA). Covidien polypropylene specimen containers (120 mL) were purchased from Fisher Scientific (MA, USA).

### Additive solution preparation

Deionized (DI) water (resistivity R = 18.2 MΩ) produced by a deionizing water system (METTLER TOLEDO, OH, USA) was used. Additive solutions were freshly prepared in-house, except for the Belzer UW solution that was purchased from Bridge to Life Ltd (SC, USA) (Fig. 1a). Additive solutions were then pH adjusted and filtered through a vacuum filter system with a 0.22 μm polyethersulfone membrane (25–227, Genesee Scientific, CA, USA). Final pH and osmolarity levels were measured using a portable pH meter (PH8500, Apera Instruments, OH, USA) and an automated osmometer (2084, Precision Systems Inc., MA, USA) (Fig. 1b). The osmolarity levels for E-Sol 5, AS-7, CPDA-1, and SAGM were measured as 270, 203, 498, and 350 mOsm/kg, respectively. For the antioxidant experiments, the dissolved antioxidants were added to E-Sol 5 at desired concentrations.

### Blood collection and processing

Packed human RBCs in CPDA-1 (~180 mL) from healthy donors were purchased from Research Blood Components, LLC (MA, USA). The company obtained informed consent from all the participants. Packed RBC units were received fresh, stored at +4 °C until a blood test was received, and used within a week. For a single pool, 3 packed RBC units were pooled in a sterile specimen container (Fig. 1c). Pooling was performed to reduce the impact of any donor-specific factors such as age, ethnicity, gender, and lifestyle and investigate only the outcomes of storage independent of donor variables. Pooled RBCs were then split at a volume of 20 mL into 50 mL conical tubes and suspended in 15 mL of an additive solution. The samples were centrifuged 3 times at 1500 g and

washed with the corresponding additive solutions before supercooling (Fig. 1d). After the RBCs were resuspended in their respective additive solutions at ~50% hematocrit levels, they were kept at +4 °C overnight. The next day, the samples were centrifuged once at 1500 g and washed with the corresponding additive solutions, and day 1 levels for all the assays were assessed. Also, the samples were aliquoted as 1 mL units in 5 mL polystyrene tubes and sealed by 0.5 mL mineral oil. The packed RBCs delivered to us were in standard PVC-DEHP bags filled with 180 mL of the samples. These bags were gas permeable for buffering purposes during storage. The polystyrene tubes used in our experiments were free of heterogenous nucleation sites and gas impermeable. Also, due to the sealing with mineral oil, the samples were not in contact with the air within the tube. The volume and surface area that the RBC samples occupied in each tube was 5000 mm$^3$ and 334 mm$^2$, resulting in a volume to surface area ratio of 15 mm. These units were stored for required periods in portable temperature-controlled freezers (Engel MHD-13, Engel, FL, USA) located in cold (+4 °C) to minimize temperature fluctuations (Fig. 1e). The temperature levels cited in the manuscript are the designated target temperature for each experimental group, and actual temperatures deviated +/−1 °C of the target temperature for short periods of time based on periodic assessment of the digital readout provided by the coolers.

### RBC quality assays

Hematological indices (mean cell hemoglobin – MCH, mean cell volume – MCV and mean cell hemoglobin concentration – MCHC) were measured on an automated hematology analyzer (XP-300, Sysmex, IL, USA). Lactate levels were measured using a blood gas analyzer (RAPIDPoint 500, Siemens, PA, USA). Measured lactate levels were normalized to the fraction of unlysed RBCs ((100%-%hemolysis)/100). Hemolysis was optically evaluated through absorbance at 540 nm with a Drabkin's-based method[58] to determine the ratio of supernatant to total hemoglobin by a plate reader (SpectraMax iD3, Molecular Devices, CA, USA) with a correction for hematocrit. Drabkin's reagent was prepared in-house. Lipid peroxidation was quantified by a Thiobarbituric Acid Reactive Substances (TBARS) assay kit (700870, Cayman Chemical, MI, USA). Malondialdehyde (MDA), a byproduct of lipid peroxidation, was quantified from the supernatant through a controlled reaction with thiobarbituric acid (TBA). The absorbance of the MDA-TBA adduct, formed by the reaction of MDA and TBA at +100 °C, was measured at 535 nm using the plate reader. Total antioxidant capacity (TAC) was quantified per μL of RBC pellet at 750 nm by an assay kit (709001, Cayman Chemical, MA, USA) based on the ability of antioxidants in the sample to inhibit the oxidation of 2,2'-Azino-di-(3-ethylbenzthiazoline sulphonate) by metmyoglobin.

### Statistics and reproducibility

Experiments were performed from 3 independent biological replicates of 3 ABO-compatible pooled RBC units (N = 3) and three technical replicates (n = 3, n = 1–3 for lactate, n = 2 for TAS and TBARS) for each biological replicate. Statistical analyses were performed using GraphPad Prism 9.1 (MA, USA). Shapiro-Wilk test was run to assess data normality. For the plots in Figs. 2, 3, and 4, a two-way repeated measures analysis of variance (ANOVA) with the Geisser-Greenhouse correction followed by Tukey's *post hoc* test was performed to evaluate significant differences between conditions: *$p < 0.05$; **$p < 0.01$; ***$p < 0.001$; ****$p < 0.0001$. For the plots in Fig. 5, a one-way analysis of variance (ANOVA) followed by Dunnett's *post hoc* test was performed to evaluate significant differences between conditions: *$p < 0.05$; **$p < 0.01$; ***$p < 0.001$; ****$p < 0.0001$. For all cases, family-wise alpha threshold and confidence level was chosen as 0.05 (95% confidence interval).

### Data availability

All relevant data are available from the corresponding authors upon request. The source data behind the graphs in the paper can be found in Supplementary Data 1.

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

## Acknowledgements

This work was supported by grants from the National Institute of Health (NIH R01HL145031 for Z.I., N.W., R.S., L.B., M.L.Y., J.P.A., O.B.U. and NIH R21GM136002 for M.L.Y. and O.B.U.) and National Science Foundation (NSF EEC-1941543 for Z.I., C.W., E.C., M.L.Y., O.B.U.). R.S. thanks the Netherland-American Foundation for the intern scholarship. We also gratefully acknowledge the support and use of facilities at the Morphology and Imaging Shared Facility and Regenerative Medicine Shared Facility provided at the Shriners Hospital for Children – Boston. The content of this article is the responsibility of the authors and does not necessarily represent the official views of the NIH or the NSF.

## Author contributions

Ziya Isiksacan, PhD (Conceptualization: Lead; Formal analysis: Lead; Investigation: Lead; Methodology: Lead; Validation: Lead; Writing – original draft: Lead; Writing – review & editing: Lead). Nishaka William, MS (Conceptualization: Supporting; Formal analysis: Supporting; Investigation: Supporting; Methodology: Supporting; Validation: Supporting; Writing – review & editing: Supporting). Rahime Senturk, BS (Investigation: Supporting; Methodology: Supporting; Validation: Supporting; Writing – review & editing: Supporting). Luke Boudreau, BS (Investigation: Supporting; Methodology: Supporting; Validation: Supporting; Writing – review & editing: Supporting). Celine Wooning (Investigation: Supporting; Methodology: Supporting; Validation: Supporting; Writing – review & editing: Supporting). Emily Castellanos (Investigation: Supporting; Methodology: Supporting; Validation: Supporting; Writing – review & editing: Supporting). Salih Isiksacan (Investigation: Supporting; Methodology: Supporting; Validation: Supporting; Writing – review & editing: Supporting). Martin L. Yarmush, MD, PhD (Conceptualization: Equal; Funding acquisition: Equal; Methodology: Equal; Supervision: Supporting; Resources: Equal; Writing – review & editing: Supporting). Jason P. Acker, PhD (Conceptualization: Equal; Funding acquisition: Supporting; Methodology: Equal; Project administration: Supporting; Supervision: Equal; Resources: Supporting; Writing – original draft: Equal; Writing – review & editing: Equal). Osman Berk USTA, PhD (Conceptualization: Equal; Formal analysis: Supporting; Funding acquisition: Lead; Methodology: Equal; Project administration: Lead; Supervision: Lead; Resources: Lead; Writing – original draft: Equal; Writing – review & editing: Equal).

## Competing interests

The authors declare no competing interests.
