## [Peer Review File · Communications Biology]

Reviewers' comments:

Reviewer #1 (Remarks to the Author):

The work by Isiksacan et al., reports the findings covering the outcomes of long-term supercooled storage of the RBCs in a selection of commercially-available additive solutions: SAGM, CPD-A1, PAG3M, SOLX (AS-7), Erythro-Sol 5 and University of Wisconsin solution typically used for hypothermic storage of tissues. Further on, hypothermic/supercooled storage was performed in E-Sol 5 supplemented with antioxidants: resveratrol, melatonin, serotonin and Trolox, in order to assess whether it could alleviate the detrimental influence of oxidative stress on the stored cells. Throughout the storage period lasting up to 23 weeks, the authors compare the rate of haemolysis, accumulation of lactate, RBC indices: MCH, MCV and MCHC, lipid peroxidation and total antioxidant capacity in different timepoints of storage in +4, -5 and -8 degrees Celsius. The authors conclude that cells subjected to supercooled storage suffered the lowest haemolysis when stored in E-Sol 5. Supercooled storage has been shown to have an advantage over traditional, hypothermic after 21 weeks of storage. Supplementation of E-Sol 5 with antioxidants enables to reduce the haemolysis levels during 10 weeks of supercooled storage. In general the work is consistent and refers to a clinically relevant problem - the attempt to increase the storage time and quality of life-saving red blood cell concentrates to be used in transfusion medicine. The authors propose a novel approach accommodating supercooling of pRBC that from the biological point of view seems rational. The introduction is neatly written and in a concise way summarizes the background for the study. However, the description of the experimental part of the study raises some concerns and requires amendments.

Major:

- 1) The main concern of the Reviewer is small N =3 (in case of lactate level measurements N= 1-3,). Human blood is a material that is relatively easy to obtain in the developed countries and packed red blood cell-based experiments do not require consent of Bioethical Committee. Moreover, artificial increase in N by so-called “technical replicates” questions the credibility of the obtained results.
- 2) The authors did not disclose any data about the donors- their sex, age or ABO status, while these factors influence the performance of the RBCs during prolonged storage. Moreover, it is not clear why the blood was pooled. Pooling of samples is performed when the volume of an individual sample may not be sufficient for testing, what does not seem to be the case here. Did the authors wanted to erase any differences stemming from different donor background? Actually, even if the reported N =3, since the samples were pooled, these are all technical replicates of the same pooled sample. All in all, the outcomes of the study, although very interesting, would be more solid if the experiments were conducted in more numerous, unpooled samples .
- 3) Was the data normality assessed before ANOVA test was run? If so, the information about the test being used should be included in the ‘Methods’ section. Moreover, the ‘Statistical analyses’ part is insufficient and needs expanding.
- 4) If the storage media were chloride free, there is no possibility of the occurrence of chloride shift, since this phenomenon the exchange of intracellular bicarbonate for extracellular chloride across the cell membrane. Thus, this conclusion seems improper.
- 5) In general, the description of results is written in an ambiguous manner, and in some places

makes the reader draw false conclusions.

6) The study shows that supercooled storage in -5C evokes a higher level of haemolysis than that present in +4C, both during standard (6 weeks) or prolonged (10 weeks) storage time. This clearly shows that supercooled storage is not a better alternative. Only after 21 weeks of storage the situation seems to be reversed, but first of all, the haemolysis exceeds the FDA-approved 1% level, and second – the results were obtained with sample N=2.

7) It would make the take-home message clearer, if conclusions of the study comprised the following: After 6 weeks of pRBC storage in E-sol 5, haemolysis measured in control samples is lower at +4 than in the -5 C. Supercooled storage in E-sol 5 haemoglobin levels exceed FDA-approved 1% level of haemolysis. Although serotonin, melatonin and Trolox were able to significantly reduce haemolysis during 6 weeks, after 10 weeks of storage, none of the applied antioxidants was able to reduce the haemolysis level below 1%.

Minor:

1) Initial assessment lasting 4 weeks in both hypothermic and supercooled conditions was appropriate, leading to the elimination of PAG3M and UW from further testing due to poor outcomes regarding the levels of haemolysis. Thus, Supplementary Figure 1 should become a part of the main body of the manuscript.

2) The following assessment of hypothermic vs supercooled storage in different media throughout 6 vs 10 weeks. The authors should comment on the fact that after 6 weeks of storage E-sol 5 was the best performing additive solution, at least in the context of haemolysis. Every panel of Figures 2, 5 and Supplementary Figure 2 should have a headline stating the storage time. Moreover, putting result descriptions, e.g. “The samples stored in SAGM showed an increased MCHC at both temperatures.” in figure captions unnecessarily clutters the content.

3) Lines 165-166 state that “Similarly, haemolysis for the samples stored in E-Sol 5 at -5 °C was significantly lower compared to all the other solutions (Fig. 2b)” but the comparison on the graph shows the difference between E-sol 5 in +4C and other solutions in -5C. This must be corrected.

4) Please remove the bar from descriptions of timepoints, i.e. ‘week-21’ etc throughout the manuscript body.

5) The authors should introduce the information explaining that the mean initial levels of haemolysis, lactate, etc. were assessed on day 1 of storage, both into methods section and figure captions.

6) Line 175- ‘mean initial lactate concentration’ instead of ‘lactate concentration’.

7) Line 190-193- the Authors show initial level of lactate and the levels measured after 6 weeks. If they want to state about ‘gradual’ accumulation, they should present data from in-between timepoints, i.e. 2 weeks, 4 weeks, etc.

8) Line 198- the Authors refer to ‘pH levels’ that are not presented to the reader and it is not clear what exactly do they refer to. Initial pH of additive solutions? This matter should be clarified.

9) The graph presenting TBARS change after 6 weeks of storage should be a part of the main manuscript, not the supplement.

10) Line 222- The use of the phrase ‘The preceding study’ suggests that these results were obtained previously in some other set of experiments, while it seems that this refers to the data presented in Figure 2. Please rephrase.

11) Line 229-230- statistical significance is not something to be measured, it can either be calculated or observed. Please rephrase.

- 12) Line 228-230 this description is misleading, suggesting there should be a comparison between +4 / -5 C haemolysis level of week 7 and week 10. Please rephrase.
- 13) Line 249 – the expression ‘tease out’ is not suitable to be used in the scientific description. Please rephrase.
- 14) Data presented in Figure 4 are not very reliable, since they are derived from N=2m propped up by technical replicates.
- 15) Figure 5- dashed lines should be clearly labelled on the graph to make the distinction between them easier.

Reviewer #2 (Remarks to the Author):

General Comment:

The authors report on long-term supercooled (-5C) storage of human RBCs at physiological hematocrit for up to 23 weeks, utilizing a commercially formulated storage solution (Erythro-Sol 5) supplemented with exogenous antioxidants, resveratrol, serotonin, melatonin, and Trolox. The technology is of interest and the analyses are relevant although the paper fails to acknowledge the increase in hemolysis for supercooled storage at every timepoint and for every storage solution. Specific methodologic concerns are listed below.

Specific Comments:

1. Introduction – while prior work employed trehalose and PEG as cryoprotectants and to inhibit ice formation – it is not clear what steps were taken to prevent ice formation in this paper. Please address this question specifically.
2. Introduction – a complete table with all components/concentrations of the various storage solutions tested would greatly enhance readability.
3. Figure 1 – please review the X-axis units (mOsm/kg) – mOsm/L is a more conventional notation.
4. Methods – please describe the specific ‘cooler’ used for storage – was temperature monitored? If so what was the measure of temperature stability during storage?
5. Methods – the volume and volume/surface area and gas permeability of the container of RBC concentrates is known to influence storage lesion accrual. Please describe each specifically and compare to that for conventional RBC storage bags.
6. Results – Figure 2C – there is a significant difference (~ 4X) in lactate generation across storage solutions; notably this difference is inversely proportional to the degree of hemolysis in Figure 2a, suggesting that the data in Figure 2C should be indexed to RBC# (lactate generation ceases upon hemolysis) to remove the bias associated with differing RBC abundance over time between storage solutions (and at differing temperatures as well). This comment also applies to Figure 3a/b.
7. The conclusion of ‘improved’ RBC storage at supercooled temperatures relative to hypothermic storage is counter to the increase in hemolysis for supercooled storage at every timepoint and for every storage solution – the text needs to be revised to reflect this simple and obvious fact.

RESPONSE LETTER

Dear Reviewers,

We thank you both for the insightful comments, critical questions, and very helpful suggestions. We believe that through your helpful comments and in this revision, our manuscript has become richer, more detailed, and clearer. In the following point-by-point response, we first introduce the question/suggestion/comment of each reviewer in *black italic non-bolded text* and then present our response in blue bolded text and new additions in **red bolded text**.

We are deeply grateful for the effort and attention to detail by both reviewers and we welcome additional feedback should you have any further questions.

Best Regards,

Dr. Jason P. Acker and Dr. O. Berk Usta

PS: We include two versions of the main text: a) one which marks all the changes made to the text (marked black for unchanged, blue for deletions, red for insertions), and b) one which shows the final version (without tracked changes) of the main text. Line numbers cited in this response letter refer to those from the final version (without tracked changes).

Reviewer #1 (Remarks to the Author):

The work by Isiksacan et al., reports the findings covering the outcomes of long-term supercooled storage of the RBCs in a selection of commercially-available additive solutions: SAGM, CPD-A1, PAG3M, SOLX (AS-7), Erythro-Sol 5 and University of Wisconsin solution typically used for hypothermic storage of tissues. Further on, hypothermic/supercooled storage was performed in E-Sol 5 supplemented with antioxidants: resveratrol, melatonin, serotonin and Trolox, in order to assess whether it could alleviate the detrimental influence of oxidative stress on the stored cells. Throughout the storage period lasting up to 23 weeks, the authors compare the rate of haemolysis, accumulation of lactate, RBC indices: MCH, MCV and MCHC, lipid peroxidation and total antioxidant capacity in different timepoints of storage in +4, -5 and -8 degrees Celsius. The authors conclude that cells subjected to supercooled storage suffered the lowest haemolysis when stored in E-Sol 5. Supercooled storage has been shown to have an advantage over traditional, hypothermic after 21 weeks of storage. Supplementation of E-Sol 5 with antioxidants enables to reduce the haemolysis levels during 10 weeks of supercooled storage. In general the work is consistent and refers to a clinically relevant problem - the attempt to increase the storage time and quality of life-saving red blood cell concentrates to be used in transfusion medicine. The authors propose a novel approach accommodating supercooling of pRBC that from the biological point of view seems rational. The introduction is neatly written and in a concise way summarizes the

background for the study. However, the description of the experimental part of the study raises some concerns and requires amendments.

We thank Reviewer 1 for all their constructive comments which helped us to substantially improve the presentation and clarify the the manuscript. We hope that the revised version of the manuscript addresses the concerns raised by the reviewer.

Major:

1) The main concern of the Reviewer is small $N = 3$ (in case of lactate level measurements $N = 1-3$). Human blood is a material that is relatively easy to obtain in the developed countries and packed red blood cell-based experiments do not require consent of Bioethical Committee. Moreover, artificial increase in N by so-called “technical replicates” questions the credibility of the obtained results.

We thank the reviewer for this comment and for allowing us to clarify.

To better explain the experimental protocol, we refer the reviewer to the image below. In this study, we recruited a total of 9 donors. We separated these 9 donors into groups of 3 based on their blood types. We pooled three donor samples of each group into a sample container. The reason for pooling was to reduce the impact of any donor-specific differences (gender, race, etc.) and observe the outcomes of supercooled storage on RBCs independent of donor variabilities. At this stage, we had 3 biologically distinct samples, namely biological replicates ($N=3$), each coming from a group of 3 donors. Then, for each experimental group (for example, storage in E-Sol 5 at $-5\text{ }^{\circ}\text{C}$ for 6 weeks OR storage in SAGM at $+4\text{ }^{\circ}\text{C}$ for 10 weeks, etc.), we aliquoted $n=3$ of 1 mL units from each biologically distinct sample. These 1 mL units of $n=3$ accounted for our technical replicates for a particular biologically distinct sample for a particular experimental group. In total, we had $n=9$ of 1 mL units for a particular experimental group. These 9 units were arbitrarily located within the cooler at the desired temperature so that we erased any potential temperature fluctuations within the cooler. When we needed to measure a parameter (hemolysis, MCHC) for an experimental group, we processed all the $n=9$ units. This way, the statistical power of our measurement was such that we reduce any inter-donor variabilities in pooled biological samples, any temperature fluctuations within the coolers, and any potential sample handling issues.

The reviewer pointed out that we had 1 to 3 biological replicates (N=1-3) for lactate measurements. This is a misunderstanding. For the lactate measurements, we had 3 biological replicates (N=3) and 3 technical replicates (n=3) for each biological replicate. In some cases, lactate levels could not be measured by the analyzer (max. measurable level was 30 mmol/L) we used, such that we could only measure 1 or 2 technical replicates for each biological replicate. To improve clarity for the reader, the image above is also included in the supplementary file as Supplementary Figure 2.

As discussed, for every measure with the exceptions noted in the manuscript, we had N=3 biological replicates and n=3 technical replicates for each. We are of the opinion that technical replication was necessary to demonstrate the repeatability of the result within the same group, and we are of the opinion that it increases the credibility of the results.

Regretfully, a clear consensus on “how many biological replicates should be performed”, “how many technical replicates are necessary”, and “how many donor samples should be pooled” in red blood cell preservation studies does not exist. We list some examples from the studies we cited in our work below:

- Reference #11: N=4, n=1, pooling: 5 donors, total donors: 20;
- Reference #32: N=3, n=1, pooling: 5 donors, total donors: 15;
- Reference #35: N=10, n=1, pooling: 0 donors, total donors: 10;
- Reference #52: N=1, n=3, pooling: 10 donors, total donor: 10.

In our case, N=3, n=3, pooling: 3 donors, total donors: 9, which was comparable to the studies in the field.

2) The authors did not disclose any data about the donors- their sex, age or ABO status, while

these factors influence the performance of the RBCs during prolonged storage. Moreover, it is not clear why the blood was pooled. Pooling of samples is performed when the volume of an individual sample may not be sufficient for testing, what does not seem to be the case here. Did the authors wanted to erase any differences stemming from different donor background? Actually, even if the reported $N=3$, since the samples were pooled, these are all technical replicates of the same pooled sample. All in all, the outcomes of the study, although very interesting, would be more solid if the experiments were conducted in more numerous, unpooled samples.

We thank the author for this comment. We agree that the donor-specific factors can significantly affect the RBCs during storage. In this study, we wanted to reduce the influence of donor-specific factors by pooling the samples, and we investigated the effect of different additive solutions during supercooled storage on the RBCs independent of the potential donor-specific factors. In a follow-up study where we perform metabolomics analysis on supercooled RBCs, we indeed focused on unpooled samples, the manuscript for which is also in preparation.

3) Was the data normality assessed before ANOVA test was run? If so, the information about the test being used should be included in the 'Methods' section. Moreover, the 'Statistical analyses' part is insufficient and needs expanding.

We thank the author for this comment. The data normality was assessed using the Shapiro-Wilk test before running the ANOVA test. We extended the “statistical analysis” discussion as suggested. Please see the main text and below for the changes:

(Methods)

Statistical analyses: Each experiment was performed from 3 independent biological replicates of 3 ABO-compatible pooled RBC units. Statistical analyses were performed using GraphPad Prism 9.1 (MA, USA). **Shapiro-Wilk test was run to assess data normality. For the plots in Fig. 2, 3, and 4, a two-way repeated measures analysis of variance (ANOVA) with the Geisser-Greenhouse correction followed by Tukey's *post hoc* test was performed to evaluate significant differences between conditions: * $p < 0.05$; ** $p < 0.01$; *** $p < 0.001$; **** $p < 0.0001$. For the plots in Fig. 5, a one-way analysis of variance (ANOVA) followed by Dunnett's *post hoc* test was performed to evaluate significant differences between conditions: * $p < 0.05$; ** $p < 0.01$; *** $p < 0.001$; **** $p < 0.0001$. For all cases, family-wise alpha threshold and confidence level was chosen as 0.05 (95% confidence interval).**

4) If the storage media were chloride free, there is no possibility of the occurrence of chloride shift, since this phenomenon the exchange of intracellular bicarbonate for extracellular chloride across the cell membrane. Thus, this conclusion seems improper.

We thank the reviewer for this opportunity to clarify our discussion.

According to the literature, chloride shift in stored red blood cells occurs when the additive solution does not contain chloride. In such cases, the intracellular chloride is transported into the additive solution. In return, the available hydroxyl ions diffuse into the RBCs:

In a pioneering study by Meryman et al., it was demonstrated that washing red blood cells with solutions without any cell permeable anions results in the loss of intracellular chloride and the counterflow of hydroxyl ions. This exchange eventually results in the increase in the intracellular pH. [Meryman HT, Hornblower M. Manipulating red cell intra- and extracellular pH by washing. Vox Sang 60, 99-104 (1991).]

A study by Lagerberg et al. demonstrated that when stored in chloride-free additive solutions, red blood cells significantly lose their intracellular chloride especially within the first 7 days. [Lagerberg, J. W., Korsten, H., Van Der Meer, P. F. & De Korte, D. Prevention of red cell storage lesion: comparison of five different additive solutions. Blood Transfus. 15, 456 (2017).]

Accordingly, we made the following revision in the sentence where we first introduce the chloride shift such that it is clear that chloride shift is from the cells into the additive solution.

(Introduction)

Additionally, they do not contain chloride, which helps promote chloride efflux from the RBCs into the solution especially in the first 7 days of storage and counterflow of hydroxyl anions from the solution into the RBCs (chloride shift) and promotes an alkaline intracellular pH during the storage.

5) In general, the description of results is written in an ambiguous manner, and in some places makes the reader draw false conclusions.

We thank the reviewer for this comment and for allowing us to improve clarity. We believe that the manuscript is now much clearer where we addressed the insightful comments by the two reviewers. We have also had the manuscript read by multiple colleagues to avoid any other ambiguity.

6) The study shows that supercooled storage in -5C evokes a higher level of haemolysis than that present in +4C, both during standard (6 weeks) or prolonged (10 weeks) storage time. This clearly shows that supercooled storage is not a better alternative. Only after 21 weeks of storage the situation seems to be reversed, but first of all, the haemolysis exceeds the FDA-approved 1% level, and second – the results were obtained with sample N=2.

We thank the reviewer for this comment. We have indeed experimentally shown that the hemolysis levels at -5 °C were higher than at +4 °C at earlier timepoints (week 6 and week 10). Therefore, in the manuscript, we presented supercooled storage as a “recent”, “novel”, and “promising” – rather than a better – alternative at these earlier time points. The supercooled storage approach, while promising as discussed, needs further optimizations and supercooling specific injury alleviation strategies which are currently under investigation in our laboratories.

We understand and appreciate the reviewer’s point regarding the hemolysis being higher than the FDA-approved 1% and the number of biological replicates at the later time points

(week 21). Hence, we revised this sentence in the conclusion which read: “*Supercooled storage was demonstrated to be significantly better than hypothermic storage at and beyond week 21.*”

(conclusion)

While hypothermic storage led to lower hemolysis than supercooled storage at earlier time points, the supercooled samples experienced lower hemolysis than the samples stored in hypothermic conditions at and beyond week 21.

This study demonstrated that E-Sol 5 is the best commercially formulated RBC hypothermic additive solution for supercooled storage and its performance can further be enhanced by exogenous antioxidant supplementation to meet FDA-mandated criteria for transfusion of stored RBC products after 6 weeks of storage.

Further we also revised the summary in our introduction:

(introduction)

We observed that the supercooled samples experienced hemolysis levels higher than the Food and Drug Administration (FDA)-approved 1% threshold.

7) It would make the take-home message clearer, if conclusions of the study comprised the following: After 6 weeks of pRBC storage in E-sol 5, haemolysis measured in control samples is lower at +4 than in the -5 C. Supercooled storage in E-sol 5 haemoglobin levels exceed FDA-approved 1% level of haemolysis. Although serotonin, melatonin and Trolox were able to significantly reduce haemolysis during 6 weeks, after 10 weeks of storage, none of the applied antioxidants was able to reduce the haemolysis level below 1%.

We thank the reviewer for this comment. We revised the conclusion to include the reviewer’s points.

(conclusion)

We reported on improved supercooled storage of human red blood cells (RBC) at physiological hematocrit levels for up to 23 weeks as an alternative to 6 weeks of hypothermic storage. We assessed two next-generation additive solutions, E-Sol 5 and AS-7, together with two conventional counterparts, CPDA-1 and SAGM, for their suitability to provide improved and extended supercooled storage. All the supercooled samples exceeded the Food and Drug Administration (FDA)-approved 1% hemolysis level standardized for up to 6 weeks of hypothermic storage. Yet, compared to the other solutions, the samples stored in E-Sol 5 experienced lower hemolysis and ensured continued glycolytic activity during storage at -5 °C. While hypothermic storage led to lower hemolysis than supercooled storage at earlier time points, the supercooled samples experienced lower hemolysis than the samples stored in hypothermic conditions at and beyond week 21. Oxidative stress was an important mechanism of injury at subzero temperatures. To this end, we recruited four exogenous antioxidants, resveratrol, serotonin, melatonin, and Trolox, to decrease hemolysis levels at -5 °C. Through the supplementation of these antioxidants in E-Sol 5, we kept the hemolysis levels significantly lower than the antioxidant-unsupplemented samples

during 10 weeks of supercooled storage. **Supplementation of serotonin, melatonin, or Trolox in E-Sol 5 kept the hemolysis below the FDA-approved level during the first 6 weeks of supercooled storage.** This study demonstrated that E-Sol 5 is the best commercially formulated RBC hypothermic additive solution for supercooled storage and its performance can further be enhanced by exogenous antioxidant supplementation to meet FDA-mandated criteria for transfusion of stored RBC products after 6 weeks of storage.

Minor:

1) Initial assessment lasting 4 weeks in both hypothermic and supercooled conditions was appropriate, leading to the elimination of PAG3M and UW from further testing due to poor outcomes regarding the levels of haemolysis. Thus, Supplementary Figure 1 should become a part of the main body of the manuscript.

We thank you for this suggestion. We had investigated UW and PAG3M in detail in our previous publications [William et al. Front. Phys. 14, 1165330, 2023]. Accordingly, here we preferred to keep this figure as supplementary information such that we can focus on the other additive solutions in the main text. Further, supplementary Figure 1 is meant to be a screening experiment which was performed using one biological replicate obtained from a single donor.

2) The following assessment of hypothermic vs supercooled storage in different media throughout 6 vs 10 weeks. The authors should comment on the fact that after 6 weeks of storage E-sol 5 was the best performing additive solution, at least in the context of haemolysis.

We thank the reviewer for this suggestion, and we have revised accordingly:

After 6 weeks of storage, E-Sol 5 was the best performing additive solution in terms of hemolysis at both temperatures. From week 6 to week 10, hemolysis levels increased in each group.

Every panel of Figures 2, 5 and Supplementary Figure 2 should have a headline stating the storage time.

We thank the reviewer this suggestion. We have revised the figure captions accordingly.

Moreover, putting result descriptions, e.g. “The samples stored in SAGM showed an increased MCHC at both temperatures.” in figure captions unnecessarily clutters the content.

We thank the reviewer for this comment and the suggestion. The somewhat lengthy descriptions in our captions stem from our desire to minimize ambiguity, improve completeness, and increase clarity for those readers that are skimming using figures only. Still, following the reviewer’s comment, we minimized as much as possible such lengthy descriptions in the captions.

3) Lines 165-166 state that “Similarly, haemolysis for the samples stored in E-Sol 5 at -5 °C was

significantly lower compared to all the other solutions (Fig. 2b)'' but the comparison on the graph shows the difference between E-sol 5 in +4C and other solutions in -5C. This must be corrected.

We thank the reviewer for pointing out the mistake. We have corrected Fig. 2b as follows:

4) Please remove the bar from descriptions of timepoints, i.e. 'week-21' etc throughout the manuscript body.

We thank the reviewer for this suggestion and we have revised accordingly to remove the bar used in timepoint descriptors.

5) The authors should introduce the information explaining that the mean initial levels of haemolysis, lactate, etc. were assessed on day 1 of storage, both into methods section and figure captions.

We thank the reviewer for this comment. We have now explained in more detail how we prepared the samples and how we measured the initial levels.

(Methods)

After the RBCs were resuspended in their respective additive solutions at ~50% hematocrit levels, they were kept at +4 °C overnight. The next day, the samples were centrifuged once at 1,500 g and washed with the corresponding additive solutions, and day 1 levels for all the assays were assessed.

6) Line 175- 'mean initial lactate concentration' instead of 'lactate concentration'.

We thank the reviewer for this suggestion, and we have revised accordingly.

7) Line 190-193- the Authors show initial level of lactate and the levels measured after 6 weeks. If they want to state about 'gradual' accumulation, they should present data from in-between timepoints, i.e. 2 weeks, 4 weeks, etc.

We thank the reviewer for this comment. The adjective “gradual” was removed from the sentence.

8) *Line 198- the Authors refer to ‘pH levels’ that are not presented to the reader and it is not clear what exactly do they refer to. Initial pH of additive solutions? This matter should be clarified.*

We thank the reviewer for this comment and the opportunity to clarify. We are referring to the initial pH levels which is measured for all solutions. We have revised the text as follows:

Hence, neither TBARS nor **the initial pH (Fig. 1b)** levels **of the solutions** alone explained why the samples stored in SAGM experienced the highest hemolysis at -5 °C (**Fig. 2a and 2b**).

9) *The graph presenting TBARS change after 6 weeks of storage should be a part of the main manuscript, not the supplement.*

We thank the reviewer for this suggestion. TBARS graph was moved into Fig. 2.

10) *Line 222- The use of the phrase ‘The preceding study’ suggests that these results were obtained previously in some other set of experiments, while it seems that this refers to the data presented in Figure 2. Please rephrase.*

We thank the reviewer for this suggestion. We have revised the sentence as follows:

It was shown in Fig. 2b that the samples stored in E-Sol 5 resulted in the lowest hemolysis during 10 weeks of storage at -5 °C.

11) *Line 229-230- statistical significance is not something to be measured, it can either be calculated or observed. Please rephrase.*

We thank the reviewer for this suggestion. We have revised the text and opted to use the verb “observe”.

12) *Line 228-230 this description is misleading, suggesting there should be a comparison between +4 / -5 C haemolysis level of week 7 and week 10. Please rephrase.*

We thank the reviewer for this suggestion. We have revised by adding the following:

We did not observe any statistical significance between week 7 and week 10 at +4 °C. This was also the case at -5 °C ...

13) *Line 249 – the expression ‘tease out’ is not suitable to be used in the scientific description. Please rephrase.*

We thank the reviewer for this suggestion. We have revised as follows:

We quantified lipid peroxidation (TBARS) and total antioxidant capacity (TAC) at all time points, to **understand** the oxidative stress ...

14) Data presented in Figure 4 are not very reliable, since they are derived from $N=2m$ propped up by technical replicates.

We thank the reviewer for this comment. We had discussed our thoughts regarding biological replicates – in response to Major point 1. Indeed, this experiment was performed using 2 biological replicates, each coming from a pool of 3 donors. Pooling was performed to eliminate donor-specific variations as discussed. For each biological replicate, we performed 3 technical replicates to demonstrate that our hemolysis assay can reliably measure the same value for a specific biological sample. We are confident that the data is reliable, and the observation can be reproduced by researchers following our methods.

15) Figure 5- dashed lines should be clearly labelled on the graph to make the distinction between them easier.

We thank the reviewer for this suggestion. We have revised the figures to clarify the dashed lines.

Reviewer #2 (Remarks to the Author):

General Comment:

The authors report on long-term supercooled (-5°C) storage of human RBCs at physiological hematocrit for up to 23 weeks, utilizing a commercially formulated storage solution (Erythro-Sol 5) supplemented with exogenous antioxidants, resveratrol, serotonin, melatonin, and Trolox.

The technology is of interest and the analyses are relevant although the paper fails to acknowledge the increase in hemolysis for supercooled storage at every timepoint and for every storage solution. Specific methodologic concerns are listed below.

We thank Reviewer 2 for all their constructive comments which helped us to substantially improve the presentation and clarify the manuscript. We hope that the revised version of the manuscript addresses the concerns raised by the reviewer.

Specific Comments:

1. Introduction – while prior work employed trehalose and PEG as cryoprotectants and to inhibit ice formation – it is not clear what steps were taken to prevent ice formation in this paper. Please address this question specifically.

We thank the reviewer for the opportunity to provide more detail on supercooling. In our previous work, we did not employ trehalose and PEG to inhibit ice formation. Ice formation was inhibited by ensuring that we sealed the top surface of the red blood cell suspension with an oil layer such that there was no contact between the suspension and air at their interface, the most likely site of heterogeneous ice nucleation. This physical method was developed in our 2019 study to achieve stable supercooled samples at temperatures as low as -20 °C [Huang et al. Nat Comm, 2019, 9 (1), 3201].

In this study, we used the same method to prevent heterogeneous ice nucleation and achieve ice-free supercooling, albeit at higher temperatures which also considerably reduces the probability of ice nucleation and freezing even without the surface sealing approach. The following revisions were made in the manuscript to better explain the concept of supercooling and to highlight that our surface sealing method hinders heterogeneous ice formation.

(Introduction)

A supercooled liquid exists in a metastable state where the liquid is cooled below its melting point (T_m) without forming ice crystals.⁴⁴⁻⁴⁷ In the supercooled state, ice crystallization occurs once the temperature reaches the homogenous nucleation temperature (T_h) or a heterogenous nucleation site triggers the nucleation between T_m and T_h .

This can be performed while warranting that organisms do not form ice below T_m and hence are protected from detrimental effects of ice crystallization.

The interface between the RBC suspensions and air within the sample tubes was the most likely site of heterogenous ice nucleation during supercooling. Hence, ice nucleation was hindered by sealing this interface using mineral oil (see Fig. 1e and Methods).

2. Introduction – a complete table with all components/concentrations of the various storage solutions tested would greatly enhance readability.

We thank the reviewer for this suggestion. We have added Table 1, which also includes the initial pH and osmolarity of the solutions, into the manuscript:

3. Figure 1 – please review the X-axis units (mOsm/kg) – mOsm/L is a more conventional notation.

We thank the reviewer for this comment. The osmolarity levels were measured using an automated osmometer (2084, Precision Systems Inc., MA, USA). The equipment measures the values in units of mOsm/kg. We prefer to keep the unit to avoid potential conversion errors and report directly the measured values.

4. Methods – please describe the specific ‘cooler’ used for storage – was temperature monitored? If so what was the measure of temperature stability during storage?

We thank the reviewer for this comment. We provided the specific brand and model of the coolers in the “Methods” section in the original text (Engel MHD-13, Engel, FL, USA). This information is now also cited within the results section as follows:

Samples were stored in coolers (Engel MHD-13, Engel, FL, USA) set at designated subzero temperatures.

All the coolers were operated in a cold room set at +4 °C to minimize fluctuations. Therefore, the effect of ambient temperature on the operating temperatures of the coolers was minimized. We did not monitor the temperatures of the coolers in a continuous manner during the operation in these studies, however, we have previously measured that Engel MHD-13 coolers achieve target temperatures with occasional deviations of +/-1 °C. These coolers feature a digital readout on their front panel, and we periodically assessed the temperature readings via this readout twice/thrice a day. The coolers were able to keep the internal temperature at the desired temperatures, but +/- 1 °C fluctuations were observed for short durations (< 30 min). To provide a more accurate description of temperate stability, we included the following sentence was included:

The temperature levels cited in the manuscript are the designated target temperature for each experimental group and actual temperatures deviated +/-1 °C of the target temperature for short periods of time based on periodic assessment of the digital readout provided by the coolers.

5. Methods – the volume and volume/surface area and gas permeability of the container of RBC concentrates is known to influence storage lesion accrual. Please describe each specifically and compare to that for conventional RBC storage bags.

We thank the reviewer for the comment. The following discussion was added:

(Methods)

The packed RBCs used in this study were delivered to our lab in standard PVC-DEHP bags filled with 180 mL of the samples. These bags were gas permeable for buffering purposes during storage. The polystyrene tubes used in our experiments were free of heterogenous nucleation sites and gas impermeable. Also, due to the sealing with mineral oil, the samples were not in contact with the air within the tube. The volume and surface area that the RBC samples occupied in each tube was 5000 mm³ and 334 mm², resulting in a volume to surface area ratio of 15 mm.

6. Results – Figure 2C – there is a significant difference (~ 4X) in lactate generation across storage solutions; notably this difference is inversely proportional to the degree of hemolysis in Figure 2a, suggesting that the data in Figure 2C should be indexed to RBC# (lactate generation ceases upon hemolysis) to remove the bias associated with differing RBC abundance over time between storage solutions (and at differing temperatures as well). This comment also applies to Figure 3a/b.

We thank the reviewer for the comment. Accordingly, we changed Fig. 2 and Fig. 3 to normalize the reported lactate levels. In accordance with the reviewer’s comment, we first got the hemolysis levels for each group, subtract these levels from 100% to calculate the percent of unlysed red blood cells. Then, we divided it by 100%. Finally, we divided the lactate levels by the corresponding percent unlysed red blood cells. The text and captions were also revised to point out that we represent the lactate levels corrected for hemolysis.

(Methods)

Measured lactate levels were normalized to the fraction of unlysed RBCs $((100\% - \% \text{hemolysis})/100)$.

7. The conclusion of ‘improved’ RBC storage at supercooled temperatures relative to hypothermic storage is counter to the increase in hemolysis for supercooled storage at every timepoint and for every storage solution – the text needs to be revised to reflect this simple and obvious fact.

We thank the reviewer for this comment. We first want to emphasize that hemolysis increases at all temperatures in time during storage, including the hypothermic storage. We agree that the time-dependent increase in hemolysis is more pronounced at subzero temperatures. We have revised the conclusion in accordance with these two facts:

(conclusion)

All the supercooled samples exceeded the Food and Drug Administration (FDA)-approved 1% hemolysis level standardized for up to 6 weeks of hypothermic storage. Yet, compared to the other solutions, the samples stored in E-Sol 5 experienced lower hemolysis and ensured continued glycolytic activity during storage at -5 °C. While hypothermic storage led to lower hemolysis than supercooled storage at earlier time points, the supercooled samples experienced lower hemolysis than the samples stored in hypothermic conditions at and beyond week 21.

REVIEWERS' COMMENTS:

Reviewer #2 (Remarks to the Author):

The work by Isiksacan et al., had been put through a revision by the Authors and the Reviewer has read the corrected version of the manuscript. Although most of the questions/ doubts raised by the Reviewer have been addressed in a satisfactory manner, some of them are still to be considered for further amendments.

Major:

#1 Now that the Authors have explained the design of the study in detail and included a Figure, there is no concern over the N number and the use of biological and technical replicates.

#2 The authors should include the information about the reduction of the influence of donor-specific factors by sample pooling in the 'Methods' section.

#3-7 The Reviewer agrees with provided explanations and introduced changes.

Minor:

#2 What the Reviewer had in mind by "Every panel of Figures 2, 5 and Supplementary Figure 2 should have a headline stating the storage time" was that above every graph there should be a heading stating the storage time, so the Reader doesn't have to go through a lengthy figure description. This is especially relevant in case of Figure 2, where two panels showing the results for hemolysis (a&b) are presented.

Otherwise, the Reviewer agrees with provided explanations and introduced changes.

Reviewer #3 (Remarks to the Author):

All concerns have been adequately addressed. Recommend acceptance without further revision.